# Effect of Positional Embeddedness on Tour Supplier Integration: Chinese Evidence

**Yin Chen, Yong Long * and Jianyin Bai**

School of Economics and Business Administration, Chongqing University, Chongqing 400044, China;
chenyin@cqu.edu.cn (Y.C.); 20140201030@cqu.edu.cn (J.B.)

* Correspondence: longyong@cqu.edu.cn

**Abstract:** This article identifies the positional attributes of travel agents (TAs) in the extended TA network in Chongqing (China), and further examines the impact of the travel agent's positional embeddedness (measured by three indicators: degree centrality, structural hole, and closeness centrality) on tour supplier integration (TSI). We applied the random permutations method (node-level) to test theoretical hypotheses based on a sample of 51 travel agents (TAs). It was found that the target travel agent's positional embeddedness could positively accelerate the sustainable integration of tour suppliers and the travel agent's structural hole attribute could significantly contribute to the TSI. The results indicated that a TA can strengthen control power by increasing the positional embeddedness to facilitate high efficiency supplier integration through which TAs can encourage the supplier development of sustainability (SDS) practices, and finally to upgrade the tourism supply chain (TSC) sustainability.

**Keywords:** positional embeddedness; centrality; structural hole; tour supplier integration (TSI); tourism sustainability

## 1. Introduction

As intermediaries in the tourism supply chain system, travel agents play the roles of information/resource centers downstream (sustainable supply chain management) and upstream (business-to-business marketing) simultaneously [1]. With the growing attention on sustainable tourism, tourism stakeholders (especially travel agents) have to face enormous pressure related to environmental protection and sustainability. Some researchers have investigated sustainability from the aspects of the tour supply chain [1–3], small tourism enterprises (STEs) [4], destinations [5,6], and society and the environment [7,8]. Generally, there are two perspectives of sustainability research: social sustainability on the macro level (community development, environmental management, health and safety management, and building networks) and firm sustainability on the micro level (sustainability performance/capabilities, sustainable relationships, resource sharing, and joint benefit). Among these scholars, Garay et al. (2017) found that sustainability implementation is related to the formal and informal connections, resource exchanges, and perceived usefulness of information with other stakeholders [4]. Similarly, Richards and Font (2019) stated that collaborative sustainable supply chain management (SSCM) required investment in formal relationships and more relational aspects. When the stakeholders have relative long-term relationships and meet their different operational goals, then relational mechanisms are more likely to lead to collaboration and integration [1]. Hence, we focus on inter-firm sustainable relationships to explore the resource integration mechanisms.

In the business environment of the tourism industry, the structural position of a travel agent in a network is emerging as one of the most important aspects to be considered in the research on resource integration. The network position of a travel agent (TA) represents opportunities to access social capital

(including fine-granted information, novel knowledge, specific asset/investment, markets/ channels) to develop new tour packages. In the research on network position, Gulati is the typical representative. He emphasized the differential informational advantages and control benefits actors can generate by being advantageously positioned in a network [9]. Then, in research on triad patterns in 1999, he noted that positional embeddedness captures the impact of the positions organizations occupy in the overall structure of the alliance network on their decisions about new cooperative ties [10]. He used centrality to stimulate the roles actors occupied in a system. Then, Gulati (2007) studied the joint dependence in dyadic exchange relationships and found that actors benefit from being in a position of power [11]. The difference of travel agents' positions in networks may increase the dependence asymmetry and lead to power asymmetry. Some researchers also found that unequal power among stakeholders might influence the balance of costs and benefits of SSCM [1], and the cost–benefit imbalance and low resource efficiency might undermine sustainability efforts [3,4,7,12–14]. Originally, Gulati and Martin mentioned that unequal power was related to structural positions and emphasized the firms to which it was tied [10]. Other scholars also highlighted the importance of positional embeddedness in sustainable supply chain research [15–17]. Despite a significant amount of attention, the benefits of the positional dimension in a network remain poorly understood. So, it is essential to study the impact of positional embeddedness on resource integration in order to improve the resource efficiency of SSCM.

Many tour firms are pursuing exchanging resources with multiple participants in a network rather than in an isolated dyadic relationship. Since the isolated dyadic tie is relatively vulnerable and cannot meet the personalized needs of tourists, it is essential for travel agents to integrate the multiple tour suppliers in supply networks to guarantee the quality of service and customers' satisfaction. So, a better understanding of the structure of such links in networks will be important for travel agents to achieve efficient integration of suppliers. We focus on the travel agent (TA) extended network to capture the sociogram and to explore the effect of a travel agent's positional embeddedness on the tour supplier integration in order to encourage the suppliers' sustainable practices and activities. Thus, we need to think about the following questions thoroughly. In which positions are the travel agents located? What is the configuration of the power relationships derived from interdependence among travel agents? What is the influential mechanism between the focal travel agent's positional embeddedness and the integration of tour suppliers? How should the supplier development of sustainability practices be directed through effective resource integration to contribute to tourism supply chain (TSC) sustainability?

The rest of this paper is organized as follows. Section 2 presents an overview of social embeddedness theory, the concept and dimension of positional embeddedness, and the interpretation of TSC network, and then introduces the definition of tour supplier integration (TSI). Several hypotheses on positional embeddedness and TSI are proposed in Section 3. Section 4 presents the process of sample collection and an estimation of the reliability and validity of relational data. Section 5 captures the sociogram of the whole network and outlines the research methods and analysis results. Section 6 includes a discussion of the results and managerial implications. Sections 7 and 8 summarize the conclusion and research limitation, and finally, provide several suggestions for future studies.

## 2. Theoretical Background

### 2.1. The Concept and Dimension of Positional Embeddedness (PE)

The embeddedness theory was first put forward by Hungarian scholar Polanyi in *The Great Transformation*, the intrinsic assertion of which was that the networks in which an actor is embedded shape the actor's behavior [18]. As Granovetter noted, instead of economic life being submerged in social relationships, these relationships become an epiphenomenon of the market. The formation of embeddedness could be understood as the process of increasingly being intertwined with other social networks, influenced by the crucial factors in focal firm's network, including trust and malfeasance, culture and reputation [19]. Prior research has frequently considered the effect of network ties,

particularly their patterns or structures [11,20–23]. Rowley et al. (2000) emphasized the informational value of an actor's structural position [24]. He emphasized the actor's positional attributes, which represent the differentiated opportunities to access fine-grained information (e.g., operational, technical, and financial information, explicit and implicit knowledge) in a network. So, it was essential to consider whether focal travel agents occupying superior network structural positions would gain an obvious advantage in accessing resources from further stimuli from the manner in which their capabilities were enhanced by network positions. On this viewpoint, Gulati and Martin (1999) argued that an actor's status/position is vital to construct informational advantages and control benefits [10]. Positions refer to the positional characteristics of travel agents in a TA extended network, such as how central the role of a travel agent is (related to core competitiveness/resource endowment), how close the position of a travel agent is to other central travel agents (related to the Euclidean distances), and how many structural holes a travel agent occupies (related to interdependence). Embeddedness in this paper refers to the state of dependence of a travel agent on their connected tour organizations (both private and public participants) in a TA extended network. Rooted in Gulati's theoretical framework, I combine the tourism supply chain context and define positional embeddedness as the extent to which a firm relies on its position to achieve informational advantages and to acquire control power derived from the interdependence among actors in a network. Briefly, centrality, structural hole, and closeness centrality are three typical positional attributes. Ghosh et al. (2016) used combined centrality to simulate positional embeddedness [15]. The extent of positional embeddedness can be measured by three indicators, which are degree centrality, betweenness centrality, and closeness centrality [25–27], in order to analyze the impact of each positional attribute on the resource integrating mechanism. Degree centrality can capture the central roles actors occupy in a TA extended network, irrespective of the specific alters involved in playing those roles. A high value of centrality describes the status of an actor as in a core position. It is easy for the core actor to exchange valuable resources with tied competitors [27]. Structural hole can be measured by betweenness centrality, which can stimulate the extent of indirect ties in a network and can measure the interdependence (the source of control power) among actors. We use structural hole to capture the tertius gaudeus role actors play in a tied network (literally, the third who benefits) [20,28], since actors in the structural hole positions can connect other pairs of unreachable travel agents, which makes their positions display the bridge attribute (described in Granovetter's strength of weak ties). The bridge position can be understood as the position where if the focal actor connected with two unreachable actors refuses to transmit information, the two actors cannot reach each other to exchange resources. That is, the focal actor can bridge the communication path between unreachable actors. The bridge attribute reflects the extent of dependence on the tertius gaudeus. Specifically, Fang et al. (2010) revealed that actors with a high bridge attribute could derive exclusive control of the information flow in the network [29], and could eventually achieve the hole effect proposed by Burt in the structural hole theory. So, it is reasonable to use structural hole to measure the actor's control power based on Emerson's dependence–power theoretical framework. As to the closeness to other central actors who have owned superior resources, closeness centrality is used to describe the extent to which the focal actor is not subject to other participants [25]. Notwithstanding the positional analysis is an important social network research issue, positional attributes of travel agents in a tourism supply chain network still need further understanding. When facing the complex configuration of a TA extended network, it is essential to capture the travel agent's positional characteristics and further analyze their impacts on the vertical integration of tour suppliers.

*2.2. TSC Network*

The conceptualization of the tourism supply chain (TSC) in extant literature is mostly defined from a single linked chain view [30–32] until Zhang et al. (2009) advocated diagnosing the TSC issues from a network perspective [33]. The tourism supply chain (TSC) is defined as a network of nodes (tourism organizations) engaged in different activities, including the procurement of raw materials,

fabrication of parts, assembly and subassembly of components, final assembly of end products, and delivery of finished products to customers [33]. Due to the close ties among tourism organizations, it is vital to consider the potential benefits of TSC from the angle of the whole network rather than the single linked chain perspective. Stakeholders in the tourism industry interact with each other to resolve their divergent business objectives across different operating systems. Since they are all embedded in different partial networks and mutually influenced in a whole TSC system, the analysis of a dyadic or triadic relationship should be put into a complex network [34]. Kim (2014) regarded the network as a broad set of firms that are motivated to accumulate tangible and intangible resources in the extended resource-based view. He emphasized that resources could be obtained from a wide network and participants in a network had resource accessibility for the network resources [35]. According to Kim's research, a supply chain network could be classified into four taxonomies: a supplier network (between a single supplier and its buyers), a buyer network (between a single buyer and its suppliers), a focal firm supply network (among a focal firm, suppliers, and buyers in multiple tiers), and a sector supply network (among manufacturers, suppliers in multiple tiers, and buyers in multiple tiers) [35]. A supply network (e.g., theme parks, hotels, restaurants, handicraft shops, and transportation operators) in tourism can generate various social capitals and a consumption network (tourist, intermediaries, such as the tour operator/travel agency) can generate the purchasing behavior. Building on the definitions of Burt [20], Coleman [36], Lin [37], and Kim [35], I define social capital in tourism as the sum of dominant and recessive resources derived from a network. This study presents an empirical study by illuminating stakeholders' engagement in public–private networking towards sustainability. Stakeholders need to utilize various social capitals for operation their business, and so the social network is expected to play a key role in sustainable tourism supply chain management. The network theory provides an insight for travel agents (buyers) in the downstream to analyze the integrating mechanism of social capital derived from tour suppliers (sellers) in the upstream of the TSC network. Zhang and Zhang (2018) pointed out that the networking of small tourism enterprises (STEs) is crucial to both their business operations and regional development [2], and tourists, neighbors, other STEs, and government were treated as major stakeholders in a social network by them. Following this logic, in this paper we concentrated on the travel agent (TA) extended network we designed (among focal travel agents, tour associations, research institutions, and OTA (Online Travel Agency) platforms) to explore the structure of collaborative relationships and to further analyze the influential relationship between a travel agent's positional embeddedness and the sustainable integration of tour suppliers by exposing the asymmetric interdependency among actors in the TA extended network.

*2.3. Understanding of Tour Supplier Integration (TSI)*

Apart from purchasing services from various suppliers, Travel agents have more of an emphasis on collaborating with outer tour suppliers to exchange resources, mainly through technical means (e.g., system integrating) and commercial means (e.g., price discounts) to achieve sustainable relationships with them, in order to reduce the transactional costs and improve the agility of TSC. Busse et al. (2016) found that supplier development for sustainability (SDS) is useful for buyers to shape the supply pool to mitigate potential supply chain sustainability risks [38]. Among the existing literature, some scholars have also researched two-party relationships from the marketing perspective. The impact of tour operators on destination sustainability has been highlighted [38,39] and the relationships between travel agents and other tour suppliers have also aroused wide concern [40–42]. Typically, Alamdari (2002) found that travel agents could have a great impact on customers' choices of airlines because neutral advice was more acceptable to customers and travelers have greatly relied on travel agents for travel information and tour packages for decades. Although the airlines could sell tickets directly to customers, they still rely on travel agents to attract additional passengers [40]. The integration of aviation resources and tour packages is mutually beneficial. Alamdari analyzed a single dyadic relationship and ignored the influence of the relational network. While these studies have enhanced the understanding of relational assets within the dyadic relationship between a travel agent

and their tour suppliers, it falls short of addressing the importance of the TA extended network beyond the immediate dyadic relationship. It is essential to analyze travel agents' integration of social capital derived from supply networks in a TA extended network rather than in a single dyadic relationship.

Scholars who concentrated on supply chain management (SCM) research have an agreement that the conceptualization of integration should be viewed from three perspectives: strategic, tactical, and operational [43–48]. Sustainable supply chain management (SSCM) adds sustainability into supply chain management (SCM) processes, to consider the environmental, social, and economic impacts of business activities [2,49]. SSCM requires strong supplier collaborations to fertilize creative product development and achieve relatively defendable competitive advantage, to the benefit of all partners. Richard and Font proposed that imbalances in the costs and benefits of SSCM can undermine sustainability efforts [40]. Hence, an important part of becoming more sustainable is sourcing and selling (more) sustainable products. Efficient supply–demand matching mechanisms can rapidly improve the sustainability performance of TSC through the reduction of transactional and communicational costs. One of the most important approaches is to efficiently integrate suppliers to reach the balance of cost and benefit of TSC. Then, the sustainability efforts will be continued in the whole TSC system. In previous SCM research, strategic supplier integration refers to the process of acquiring and sharing social capital, including operational, technical, and financial information and related knowledge with the supplier, and vice versa. Activities commonly associated with supplier integration include partnerships, co-development activities, joint planning meetings, and shared information systems [47]. Strategic supplier integration may prompt various tour suppliers to offer useful information that may result in the target travel agent's better product planning for growth and profitability in the long term. A travel agent–supplier relationship can be understood as a buyer–supplier collaborative relationship in a TSC network. Each can be viewed as an actor performing activities to pursue value generation [50]. Busse et al. (2016) illustrated that supplier development of sustainability (SDS) could foster cooperation between buyers and suppliers and reduce supply chain sustainability risks for buyers, and SDS practices could also improve suppliers' competitive position and financial performance [38]. Integration usually means deep inter-organizational cooperation by resource sharing, system embeddedness, and knowledge co-creation. I combine the tourism context and define the tour supplier integration (TSI) as being involved in the travel agent's sustainable collaboration with multiple tour suppliers in a TA extended network to construct the synchronized processes and targets to enhance the travel agent's control of social capital derived from supply networks, and finally to improve the supplier development of sustainability (SDS). A successful TSI can increase tour suppliers' interactions with travel agents in the development of new tourism packages through information sharing, strategic coordination, and resource recombination.

## 3. Hypotheses

A complete representation of our conceptual model is shown in Figure 1 below, on the basis of prior literature.

### 3.1. Degree Centrality and TSI

In the social network theory, the centrality index is the source of power owned by the participants occupying these positions in networks [23,26,27]. Furthermore, it is relatively the most suitable index to capture an actor's visibility, information access, and potential communication activity [51]. Freeman mentioned that when a particular person in a group is strategically located on the shortest communication path between pairs of other members, this person is in a central position [25]. It was proven that a player in a central position could own a higher status and power relationship, which means the higher the degree of centrality, the more central position a focal firm would occupy [26,27]. We used the standardized value proposed by Freeman in 1979 to index the degree centrality. If there is

a graph containing n TAs, the max number of potential ties of target TA's connecting to other TAs in a certain network will be (n-1). Then,

$$C_D(n_i) = \frac{\sum X_{ij}}{n-1}. \tag{1}$$

$X_{ij}$ is a binary number of 1 or 0, indicating whether the TA $j$ is related to $i$ or not. $n$ is the number of TAs in a TA extended network and it is crucial to standardize the calculating process by dividing by $(n-1)$, which represents a TA's largest possible sum of edges linking $n_i$ with every other actors in networks. Under the conditions, each participant is reachable from all others either directly or through $n_i$. This value is usually applied to investigate the degree of direct relationship [26], and to measure the TA's own capability and attractiveness without considering control power over others.

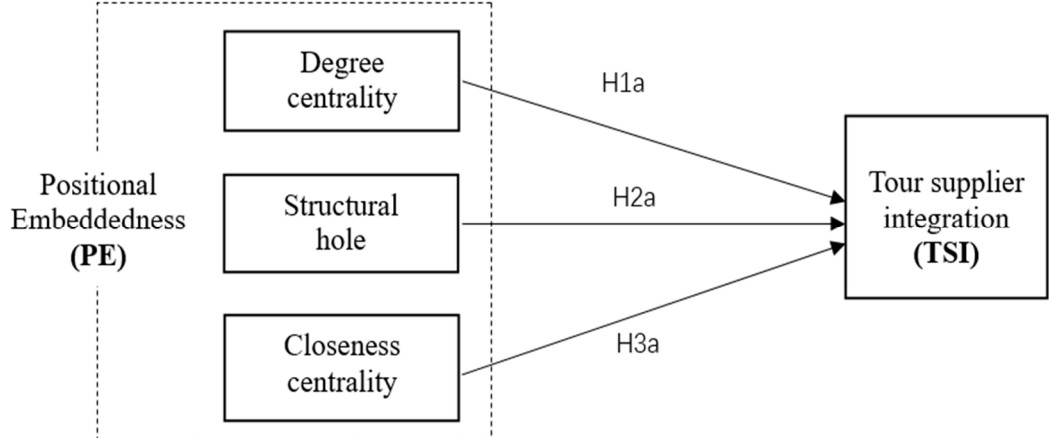

**Figure 1.** Theoretical research framework.

Generally, the reason why the actors occupying more a central position can have more informational advantages in a TA extended network is that the superior position can make the actor more visible and attractive [23]. They will have different capabilities and mechanisms to exercise control depending on the degree of centrality. More central TAs have greater power than organizations that are less central [52]. From a resource dependency perspective, TAs in a superior position can obtain more opportunities to be discovered by external partners in TSC networks and access more social capital through integrating behavior because of their high visibility and attraction. Then, other TAs will be more dependent on the central ones. That is, a high degree of centrality can increase other TAs' high dependencies on central TAs' critical resources and high dependence will ultimately enhance the degree of central TAs' control power. The information advantages resulting from degree centrality have been a recurrent core theme during the network analysis process [51]. It was found in the literature that the actors who occupy the most central positions could become the leaders, whereas the most peripheral ones play the role of the followers [53]. Similarly, Gulati (1998) stated that the more central an organization's network position, the more likely it is to access fine-grained information from a larger pool of potential tied partners in the network [9]. If a TA has greater access to information and higher visibility than other counterparts in the TA extended network, then inter-organizational ties and resource sharing should be more common among different TAs occupying similarly central positions [10], since other participants (no matter whether they are in marginal or core positions) want to strengthen their central positions by connecting with the central organization and achieving the synergistic effect of brand and reputation. Meanwhile, firms in the center of resource exchange networks are more likely to build new ties with new entrants [54]. This means a TA's high degree of centrality can contribute to improve their resource accessing capability by building new ties. Due to the central TA's informational advantages, it is easier to attract outer supplier resources embedded into the TA extended network, because a competitive supplier prefers to exchange resources with a central TA to achieve the synergistic effect of central TAs' high visibility and attraction. Then, it is more

possible to construct a sustainable buyer–supplier collaboration to reduce the transactional costs and increase the sustainability of TSC. Thus, we propose the following hypothesis.

**Hypothesis 1a:** *Travel agents' degree centrality in TA extended network enhances the positional embeddedness, and positively affects tour supplier integration (TSI).*

### 3.2. Structural Hole and TSI

A point would occupy the bridge position if it fell on the shortest path between pairs of separable points. Other members of the network are assumed to be responsive to actors in such a bridge position who could make a difference to the group by withholding information, coloring or distorting it during transmission [25]. Burt used the term structural hole to describe the relationship of non-redundancy between two contacts [20]. The travel agent occupying a structural hole is usually capable of controlling other members and deriving the network benefits from non-redundant contacts. The conception of betweenness was mainly constructed by Freeman in 1979. It was used to index the potential of a point for control by counting its opportunities for control [51]. Betweenness centrality is a useful indicator for detecting the structural hole and is regarded as an index of control power over other unreachable actors who greatly depend on the one between them to transmit information. If a travel agent (TA) fell on the shortest communicating path between other separable TAs, then the degree that it stands between others would be very high and the betweenness value would reach the maximum [26]. We assumed that the number of geodesics linking TA $n_j$ and TA $n_k$ was represented by $g_{jk}$, and $b_{jk}(i)$ measured the probability of TA $n_i$ being on the shortest path between $n_j$ and $n_k$. If $n_i$ fell on the only geodesic or all the geodesics linking $n_j$ and $n_k$, then $b_{jk}(i) = 1$. In these cases, $n_i$ could control communication because it was a necessary link between the other two TAs. If $g_{jk}(i) =$ the number of geodesics linking $n_j$ and $n_k$ that contain $n_i$, then $b_{jk}(i) = \frac{g_{jk}(i)}{g_{jk}}$ was the probability that TA fell on a randomly selected geodesic linking $n_j$ and $n_k$. So, the standardized formula of betweenness centrality is as follows:

$$C_B(n_i) = \frac{\sum\limits_{j}^{n} \sum\limits_{k}^{n} b_{jk}(i)}{(n-1)(n-2)}, \, (j \neq k \neq i, j < k). \tag{2}$$

$C_B(n_i)$ is an index of the overall partial betweenness of TA $n_i$. Whenever $n_i$ falls on the only geodesic connecting a pair of TAs, $C_B(n_i)$ is increased by 1. When there are alternative geodesics, $C_B(n_i)$ is increased in proportion to the frequency of occurrence of $n_i$ among the alternatives. This indicator actuality indexes the control capability of TA $n_i$ on others [26]. Burt argued that if focal firm A fell on the only geodesic linking other pairs of disconnected points, then it was reasonable to believe that firm A occupied the structural hole position. Since the other two points rely on A to transmit the flow of information, firm A could contact two heterogeneous information flows at the same time. Then, A might play the role of tertius gaudens to maximize their own control benefits through withholding information, coloring or distorting it during transmission. TAs in the tertius roles can create advantages for themselves by playing one off against the other and brokering tension between the other TAs. These advantages, explained as the hole effects by Burt in the structural hole theory [20], can be translated into concrete benefits in the form of favorable terms in their exchange relationships with partners [55]. Therefore, a travel agent in the role of tertius gaudens can expand their scope of ties much more easily because of the information advantages and control benefits derived from the structural hole [18,56]. The more structural holes a travel agent has occupied in a TA network, the more participants will be dependent on them. Dependence is the source of power [57]. So, the travel agent's control power will be enhanced eventually due to the increase in the other actor's dependence. In a supply network, tour suppliers are commonly in concentrated and unchangeable geographical locations. As tour resource owners, they play a dominant role in the TSC network. Compared with the

TAs with no structural holes, the central TAs occupying structural holes can build more informational advantages based on the control powers over other indirectly connected participants. Due to the comparative superior control benefits of TAs with structural holes, they can play the rule-makers to encourage sustainability practices and they are powerful at supplier integration because suppliers expect to access the complementary resource and enjoy the spillover effect to upgrade performance. In these circumstances, The TAs can enhance their own advantages by bridging structural holes and select the appropriate suppliers to exchange resources based on the sustainability performance, in order to advocate the environment, society, responsibility, and sustainability activities in TSC system. Thus, we propose the following hypothesis.

**Hypothesis 2a:** *Travel agents enhance their tour supplier integration (TSI) by bridging structural holes.*

*3.3. Closeness Centrality and TSI*

Closeness centrality describes the degree to which a focal firm is not subject to other members in a network. A higher value of closeness centrality represents a lower dependence on other participants [51]. Freeman used Euclidean distance to measure the closeness centrality. The calculating formation is as follows:

$$C_c(n_i) = \left[ \sum_{j=1}^{n} d(n_i, n_j) \right]^{-1},$$

(3)

where $d(n_i, n_j)$ represents the Euclidean distances between node $n_i$ and $n_j$. $C_c(n_i)$ is the value of closeness centrality from node $n_i$ to other nodes, and a lower value represents lower transitivity to others, which means the target node is located in a relatively peripheral position in networks. In a TA extended network, a high degree of closeness centrality indicates that the travel agency is less dependent on other participants and will be more powerful. Ford et al. (2012) found that the source of power asymmetry is the imbalance of dependency on critical resources during the tourism distribution network research [58]. A less dependent travel agency (high closeness centrality) is able to appropriate a larger portion of the total value derived from the network because they usually obtain the vital resources or control access to core capital, resources, markets, and information. They can benefit from power asymmetry through behaving more opportunistically, with little fear of being revenged by other dependent partners (e.g., violation of written contracts, failing to honor informal agreements, falsification of information, and quality shirking) [58–60]. On the other hand, independent TAs can enforce sanctions against individuals who violate shared beliefs or norms of behavior, which makes independent TAs (high closeness centrality) relatively more powerful than dependent TAs [28]. Hence, dependent TAs (less powerful) will be highly dependent on other partners' critical resources to pursue mutual strategic goals (exploiting scale economies, reaching tour package improvements) and try to balance the power asymmetry through resource integration [52]. Since their visibility and attractiveness are much lower than independent TAs (more powerful), they have to cooperate with powerful TAs to improve the closeness centrality in the network. Because of the valuable resources controlled by powerful TAs (more independent), the tour suppliers are more willing to cooperate with powerful TAs to share resources, information, and markets. Then, relatively independent partners rely on the key resources they control to attract the aggregation of internal and external network resources, so as to achieve the purpose of efficient integration. Thus, we propose the following hypothesis.

**Hypothesis 3a:** *Travel agents' closeness centrality in a TA extended network can positively contribute to the tour supplier integration (TSI).*

## 4. Research Methodology

### 4.1. Questionnaire Design

The initial questionnaire was designed in English, and then was translated into Chinese by an operational management expert in China. Taking the Chinese commercial context into consideration, we renewed the measures based on the understanding of the constructs and our observations during visits and interviews of tourism companies. Finally, we had two versions of the questionnaire. First, we submitted the questionnaire to academics and supply chain executives for their reviews and did five preliminary interviews with selected travel agents to further revise the questions. Then, we did the pilot survey on a sample of 20 (actually 18, two firms' feedback was considered invalid because of the missing information) local travel agents by face-to-face discussions, email, online contact, and telephone. After confirming the items were understandable and relevant to practice, we mailed the questionnaire. Follow-up telephone calls and e-mails were used to improve the response rate when accounting for the nonresponse from investigated firms. Out of 156 companies contacted, a total of 200 questionnaires were distributed, and we interviewed the CEO/presidents of 21 travel agents by semi-structured interviews. We tried to form the focus groups with experts from universities/associations and executives from tour firms to discuss and explore some creative definitions in the tourism industry. Finally, 102 questionnaires from 51 travel agents were valid and usable out of the returned ones during this investigation. The response rate was 12% via telephone, 28% via email, and 60% via questionnaires distributed. This paper focused on the focal travel agent's key contacts with tourism associations, governmental agencies, research institutions, and OTA platforms. After comparing the early samples with the late ones, the variation coefficients of control variables (including firm age, firm size, firm attribute, connection type, and business scope) illustrated that the differences between these variables were not significant ($p < 0.05$), which meant the samples in our study were representative and reliable.

### 4.2. Variable Measures

The operational indicators were all measured using a five-point Likert scale and the complete scales are listed in questionnaires. In order to eliminate homologous deviation, the questions related to tour supplier integration (answered by operation managers) in Appendix A were separated from the questions involved in mutual relations (answered by CEOs/presidents) in Appendix B. The focal firm's positional embeddedness was measured by centrality, Freeman's betweenness (to measure the structural holes), and closeness centrality, based on the previous literature on network research [16,24,25,28,35]. I provided a list of the elected travel agents (TAs) and related organizations in Chongqing for respondents to select the ones they connected/cooperated with frequently. Then, I constructed an adjacency matrix of inter-TA communication among TAs and related organizations based on the relational data. The matrix was symmetrized to focus on bilateral connections and to calculate positional indicators, including degree centrality, structural hole, and closeness centrality. These three indicators were applied to measure the TA's positional attributes in a TA extended network. Different from the ego network calculation, the indicators in this paper were generated from the whole network context. Liu (2004) illustrated that the calculation of a firm's positional indicators without a specific whole network context was not comparable [26]. As for the TSI, we considered the mature scales in SCI (supply chain integration) research in the manufacturing industry [61,62], and renewed some measures to better explain the characteristics of tour suppliers in the tourism industry. As summarized in the literature review, we focused on the integration of social capital derived from supply networks, which was commonly researched in the dimension of supplier integration. We included various control variables used in supply chain integration studies in a tourism context. We measured firm age as the natural log-transformed number of the difference between the investigated year and established year and measured firm size as the natural log-transformed number of employees. The region of city was set to control the regional effects on the tourist demand and tourism performance. We included a dummy variable business scope (coded 1: innovative business, such as R&D (research and development) on

technology, innovative marketing, 0: the other operational business) to control the interference of the core business type on the external integration. A dummy variable firm type was set to control for the effect of the firm's nature on external integration.

### 4.3. Sampling and Data Collection

The target respondents for this study were Chinese travel agents engaged in the tourism industry. Data were collected in the Chongqing municipality (the western city of China and governed directly by a central government). To obtain a representative sample, we used the convenient sampling approach to select travel agents located in four regions as our sampling pool, because a representative respondent with deep private relationships was more beneficial for us to collect the group data to research the whole network (the main districts of Chongqing represented by Yuzhong and Nanan, 40%; Western region, 10%; Northeast region represented by Wushan, Fuling, Wuxi, and Fengjie, 30%; Southeast region, 20%). The majority of travel agents and the related organizations in our sample had been established for more than six years, to avoid the interference coming from the differences in business experience and network relationships caused by company age. For each randomly selected travel agent, we mainly interviewed two key respondents, who had different job titles, such as supply chain manager engaged in operational system and CEO/president in charge of strategies and performance (see Table 1).

**Table 1.** Statistics of the respondents.

| Percentage of Respondents | |
|---|---|
| **Job title** | |
| CEO/president | 34.69% |
| Vice president | 6.80% |
| Director | 27.21% |
| Manager | 31.29% |
| Firm age (Years) | |
| 6–10 | 37.25% |
| 11–20 | 31.37% |
| 21–30 | 29.41% |
| >30 | 1.96% |
| Number of employees | |
| <10 | 37.25% |
| 10–50 | 58.82% |
| >50 | 3.92% |
| Firm attributes | |
| State-owned company | 7.84% |
| LLC (limited liability company) | 78.43% |
| Joint-stock company | 1.96% |
| Branch company | 7.84% |
| Individual | 3.92% |

The characteristics of the target sample of 51 travel agents were as follows. For the job titles, CEO/president and operational manager were the target respondents, so the proportions were high (34.69% and 31.29%, respectively). Then, we also interviewed the vice presidents and directors in some travel agencies due to the cross-functional systems. For firm age, there were 19 travel agents established for 6–10 years (37.25%). Sixteen travel agents had been established for 11–20 years and fifteen established for 21–30 years, respectively at the percentages of 31.37% and 29.41%. In terms of employee profile, 37.25% of the responding firms had fewer than 10 employees, 58.82% had between 10 and 50 employees, and 3.92% had more than 50 employees. Thus, it could be seen that travel agencies are asset-light service institutions and usually operate on a small scale. For the firm attribute, the largest proportion was limited liability companies (40 TAs, 78.43%). State-owned and branch companies were equally at a percentage of 7.84%, and individually-owned companies were 3.92%. Very few travel

agents belonged to joint-stock type. The responding profiles illustrated that the operations of the investigated travel agents were mostly turned to marketization. Since the majority of respondents' company policies were against sharing sensitive information, we identified several informants (15 at present) to provide the messages/gossip in the long-term, and contacted them by telephone to obtain the preliminary agreement to participate and to record the changes in the focal TA's social relationships regularly. Key contacts were identified by reviewing commercial interactions and cooperative contracts, and interested TAs were asked to provide the research team with key contact information for all strategic cooperative relationships.

## 5. The Sociogram and Analytic Results

### 5.1. The Sociogram of the Whole TA Extended Network

When dealing with the network data, if respondent A selected B as the linking firm but B did not select A, then it was reasonable to believe that the ties between the two organizations were weak. Conversely, strong ties existed between two respondents when they both selected each other at a similarly high extent. Each organization in a TSC network might be weakly connected to a different extent, no matter how directly or indirectly, because tourism is a highly integrative industry. So, we used the UCINET6.0 software to calculate the matrix of relational data (positional indicators see Appendix C). The whole TA extended network structure of investigated tour organizations in Chongqing is displayed in Figures 2 and 3. TA1–TA51 represent the 51 investigated travel agents. A1–A4 represent the Chongqing tourism bureau, Chongqing tourism association, Chongqing municipal people's government, and research institutions in sequence. S1–S4 represents four strongly-tied OTA organizations, Ctrip, Tuniu, Fliggy, and Qunar. In the process of measurement, the connections between two travel agents, travel agents and other participants such as tourism associations, and research institutions and OTA organizations were systematically taken into consideration. The number of ties was 1463 and the overall density was 0.4275, with the degree centrality of the entire network up to 25.02%. The average distance between any two TAs was 1.581 and the distance-based cohesion (compactness) was 0.712, which means the cohesiveness of the TA network was great. In the closeness of the whole network, network in-centralization (31.21%) was higher than network out-centralization (22.24%), which means that the demand for information access was greater than information sharing.

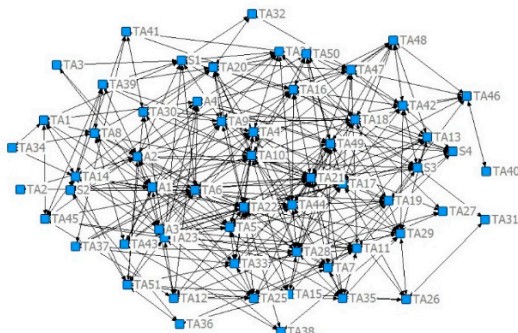

**Figure 2.** The strong ties in the TA (travel agent) extended network.

In Figure 2, travel agents (TAs) in the central area of the whole network occupy the core positions, with a high frequency of resource exchanging (including knowledge, intelligence, consultation, and privilege). These TAs as resource aggregators own the control powers over other participants, based on the multiple strong ties. The strength of the ties of central TAs with associations and OTA platforms were similarly high and the other related organizations (research institution, association/government, OTA platform) all displayed the characteristics of the resource receiver. TAs would contact with these organizations automatically to promote their social status in networks. In addition to the strong ties, the TA extended network was also fulfilled with weak ties, which made some TAs display an

obvious attribute of a structural hole in Figure 3. Such TAs had a certain influence on other embedded competitors and played roles of exchanging the center of information flow in the whole network. In summary, we can observe the strong demand for information exchange in the TA extended network.

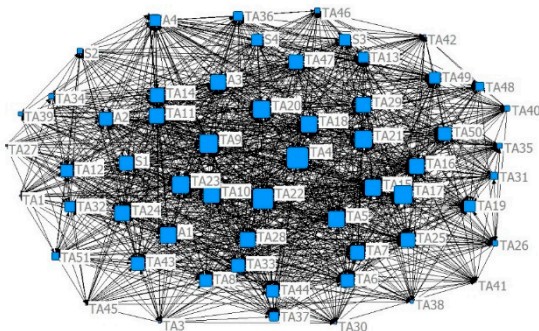

**Figure 3.** The sociogram of TA (travel agent) extended network.

*5.2. Regression and Results*

Some scholars investigated the integration of the supply chain as a unidimensional construct [22,48,63–66]. Others broke integration into multiple dimensions, in which internal and external integration were the most typical ones [43,44,47,61,62,67]. Based on the reliable scales designed by Narasimhan and Kim, Stank et al., and Flynn et al. in SCI research [43,61,62], we mainly focused on supplier integration in the tourism industry. According to the research theme, we revised the questions derived from traditional integration research and redesigned them from the perspective of an intelligence network based on the common four dimensions (emotion, intelligence, consultation, and trust) of question design in social network research. The designed questions included the degree of service capacity shared (SI1), the degree of tour product design involvement (SI2), and demand forecast shared (SI3). We added two additional indicators, except for the indicators mentioned above, including "my firm effectively shared operational information externally with selected suppliers" (e.g., shared information by stable procurement) (SI4), and "we helped a major supplier to improve its process to better meet our needs" (e.g., integrated the data systems with key suppliers) (SI5). Since the scales were originally designed in English and then interpreted into Chinese, the indicators were strictly examined in this study. First, KMO (Kaiser–Meyer–Olkin) and Bartlett's Test (KMO test is used to check the correlation and partial correlation between variables, and the value is between [0, 1]. The correlating effect is strong when the KMO value is above 0.7. Bartlett spherical test is applied to test whether the correlation matrix is a unit matrix or not. if Sig. < 0.05 ($p < 0.05$, the correlations among variables are significant and factor analysis is valid.) were applied to test the construct validity of the indicator structure (KMO > 0.7, $p < 0.001$), which proved the questionnaires were valid in structure. EFA (exploratory factor analysis) was further used to test construct unidimensionality. Some indicators with factor loading lower than 0.4 simultaneously needed to be deleted, then standardized Cronbach's alpha was applied to test the reliability of remaining indicators (see results in Table 2). The results proved the structure of questionnaire was reliable.

The node-level hypotheses test approach was applied to perform the correlation analysis and multiple regression. This method is commonly used in social network analysis to test the hypotheses that the variables involved were all attribute indicators of nodes. We can use this method to analyze the relationships among attribute variables by the multiple regression approach based on the network data. Different from the traditional OLS (ordinary least squares) regression, the logic in the node-level test is the random permutations method, by which R-square and the sampling distribution of regression coefficients are generated during the calculation progress.

It can be found from Table 3 that three positional indicators were all highly correlated with tour supplier integration (TSI) from the correlation matrix. Concretely, degree centrality (0.913), structure

hole (0.883), and closeness centrality (0.917) were positively correlated with TSI. It is rational to test hypotheses by regressing the variables.

**Table 2.** Factor analysis of integrative activities and reliabilities.

| Suppliers Factor | Integrative Activity | Std Dev |
|---|---|---|
| Factor loadings | | |
| 0.773 | The degree of service capacity shared (SI1) | 0.681 |
| 0.841 | The degree of tour product design involvement (SI2) | 0.749 |
| 0.817 | Demand forecast shared with major supplier (SI3) | 0.783 |
| 0.839 | Novel information shared by stable procurement (SI4) | 0.702 |
| 0.820 | Integrated the data systems with key suppliers (SI5) | 0.764 |
| Factor analytic and reliability statistics | | |
| 0.850 | Kaiser–Meyer–Olkin measure of sampling adequacy | |
| 117.568 | Bartlett test of sphericity | |
| 0.00000 | Significance | |
| 0.8769 | Standardized Cronbach's alpha | |
| 67.129% | Total variance explained | |
| 3.356 | Eigenvalue | |
| 0.0613 | F test | |

**Table 3.** Correlations based on randomization tests.

| Variable | Correlation | | | |
|---|---|---|---|---|
| | 1 | 2 | 3 | 4 |
| 1 Degree centrality | 1 | | | |
| 2 Structural hole | 0.836 | 1 | | |
| 3 Closeness centrality | 0.990 | 0.849 | 1 | |
| 4 Tour supplier integration (TSI) | 0.913 | 0.883 | 0.917 | 1 |

Table 4 describes that R-square (0.882) and the probability test of the regression results (0.008) in the model were relatively high, which shows the model fit was ideal. It can be seen from the correlation matrix that three positional variables were positively relevant with TSI. So, on the condition of other variables being unchangeable, the test results of the positive relationship between degree centrality and TSI were relatively significant ($\beta = 0.365$, $p = 0.287$). Hypothesis 1a was proven. Similarly, Hypothesis 2a was also proven because of the significant test results between structural hole and TSI ($\beta = 0.382$, $p = 0.099$). The test of coefficient between closeness centrality and TSI was relatively significant ($\beta = -0.231$, $p = 0.333$). Hypothesis 3a was partially supported. In conclusion, a TA's positional embeddedness was positively related to the tour supplier integration (TSI). Structural hole, as one of the three positional indicators, was the main determinant of TSI. Notwithstanding how degree centrality and closeness centrality were positively correlative with TSI, which might be aroused by random factors.

**Table 4.** Regression coefficients for TSI (Tour Supplier Integration)

| Variable | Model Fit | | | | | |
|---|---|---|---|---|---|---|
| | Standardized Coefficient | Proportion as Large | R-Square | Adjusted R-Square | One-Tailed F Value | One-Tailed Probability |
| 1 Degree centrality | 0.365 | 0.287 | | | | |
| 2 Structural hole | 0.382 * | 0.099 | 0.882 | 0.873 | 117.504 | 0.008 |
| 3 Closeness centrality | 0.231 | 0.333 | | | | |

Note: * $p < 0.1$. Permutation test is commonly applied when variables are not independent.

## 6. Discussion

In this article, we utilized the survey data of 51 travel agencies in Chongqing to explicitly examine how focal TAs' positional embeddedness could affect the efficiency of tour supplier integration. Although the testing results of degree centrality–TSI and closeness centrality–TSI were relatively significant, the testing result of structural hole–TSI was very significant. Our study is the first effort to examine the influencing mechanism between travel agents' positional embeddedness and tour supplier integration, then to extend the current exploratory research on tour supplier integration (TSI) based on the social network theory.

A focal travel agent's degree centrality and closeness centrality can enhance the extent of positional embeddedness, and can further encourage the tour supplier integration (TSI). A focal TA with a higher degree of closeness centrality can obtain a higher degree of dependence from other participants, and can seize more initiative in a resource exchange than average travel agents. It is reported in Table 2 that TA2 was in the peripheral position ($C_D(TA2) = 0.448, C_C(TA2) = 0.64$) and TA4 occupied the absolute central position ($C_D(TA4) = 0.914, C_C(TA4) = 0.92$). There were 26 nodes connected with TA2 and 53 nodes were connected with TA4, which illustrates that TA2's network size was smaller than TA4's. In the aspect of connection with counterparts, TA2 had tied 17 travel agents whilst TA4 had constructed 45 business links with counterparts. The analytic results suggest that the travel agent TA4 owned the absolute advantages on the degree centrality and closeness centrality, and it was proven in previous studies that an individual/organization with high centrality is always in the role of information/resource concentrator [9,25,56,68]. Comparing TA2 with TA4, we can notice that local government, business association, research institution, and OTA organizations were tied with both TA2 and TA4, such as the Chongqing tourism bureau, Chongqing tourism association, Chongqing municipal people's government, university, and four typical OTA organizations. Travel agent TA4 had developed close cooperative relationships with four OTA organizations, including Ctrip, Tuniu, Fliggy, and Qunar, and the managers have built some private ties with local research institutions in universities, such as Chongqing University and Chongqing Normal University. During the research process, it was found that TA4 had higher visibility and attractiveness than TA2 and a high degree of trust, prestige, and reputation.TA4 had an obvious advantage over TSI and a better effect of resource attraction and aggregation. Tour suppliers are more willing to cooperate with the travel agents who have occupied superior positions, to exchange complementary resources. Since the emotional/private connection with other participants contains the focal travel agent's trust, they are more likely to ally with each other when facing malfeasance, opportunism, and environmental uncertainty [69]. In the commercial context of China, emotional ties and information are equally important in the buyer–supplier relationships. Focal travel agents are more willing to make specific relational investments and to provide emotional assistance in consideration of the trustable and sustainable relationship with tour suppliers. Then, the information exchange and resource integration will be upgraded into a further stage.

The structural hole attribute of travel agents can enhance the extent of positional embeddedness and can positively affect the tour supplier integration (TSI). There is no doubt that TA21 was the one who had bridged the most structural holes ($C_B(TA21) = 2.343$), and most of their connections ($Ties = 987$) were indirect ties. Compared with central TA4, the sum of structural holes bridged by TA4 was less than those bridged by TA21, even though TA4 occupied the most central position ($C_B(TA4) = 1.573$). Actors need to rely on the ones bridging structural holes to transmit flows of information. These bridges represent valuable channels to ensure the fluent diffusion of information. Building the bridge position is the key to integrating novel information. Burt mentioned that the individual who occupied the bridge position could exchange resources easily with others, and could also derive control benefits from the position and own relative more resources. Our study has confirmed this view. The results illustrate that the structural hole can enhance the tour supplier integration positively. Additionally, TAs bridging structural holes may be able to access complementary resources from supply networks, may identify the trustable partners and potential allies easily, and may sense the impending threats and opportunities more quickly than others who are not so positioned in the TA extended network.

Consequently, positional embeddedness should go beyond the immediate ties of firms and emphasize the informational value of the structural position in the network. These advantages can be translated into concrete benefits in the exchange relationships between buyers and suppliers. As the concentrator of multiple heterogeneous resources, TAs with structural holes will usually care about suppliers' sustainability performance and try to improve the TSC sustainability to achieve high satisfaction through the resource integration with tour suppliers.

## 7. Concluding Remarks

This paper extended the TSC research by constructing a theoretical model of travel agents' positional embeddedness and integration of tour suppliers (TSI) based on the social network theory. In order to further empirically test the hypotheses, we investigated how the efficiency of TSI will be affected by positional embeddedness in the context of the tourism industry in Chongqing (China). A better understanding of the tour buyer–supplier relationships and the taxonomy of ties in a TA extended network may be helpful for travel agents to achieve efficient TSI and to construct a sustainable collaboration with tour suppliers. This paper has set out a new direction for TSI research. We have gained several enlightenments. First, travel agents, no matter whether they are in central or peripheral positions, should seek and develop strong ties with actors in absolute central positions to increase the degree of centrality, achieving their own business goals of improving their visibility and attractiveness. Since a high degree of positional embeddedness will help focal TAs to enhance their informational advantage, other participants will be more dependent on them. The increase in dependence will strengthen the focal TAs' control powers. Second, the central travel agents bridging structural holes may take advantage of the transmissibility of existing links to construct more effective ties and translate potential ties into actual ones to strengthen the control benefits in the network. Third, it was found that the central TAs with obvious structural holes can be more powerful when exchanging resources with suppliers, based on the critical/valuable resources they controlled. They can build the benchmark to select qualified tour suppliers to cooperate with and direct their sustainability practices to improve the sustainability of the whole TSC system (the governmental departments have proposed a high request for ecological benefits and environmental protection efficiency), as well as to satisfy customers sustainably in the long run (tourists will trust sustainability-consciousness suppliers more due to their social responsibilities). TAs as the intermediaries in TSC networks can ally suppliers with sustainability practices to integrate resources and can encourage sustainable consumption choices in markets to achieve the dual strategies and operating goals, and finally to upgrade the sustainability performance of the whole TSC system.

## 8. Limitations and Further Research

Inevitably, our study has various limitations. First, this study is limited in that the proposed model was tested using positional dimensions (degree centrality, structural hole, and closeness centrality). The importance of the characteristics of the TA's position on the integration of tour suppliers was emphasized in the position-oriented perspective. It should be interesting to test whether other structural variables can explain the variation in TSI. Second, the (supply chain) sustainability was discussed but not in depth. The measurement of sustainability performance is a worthy issue to explore. So, further studies can expand the conceptual framework by adding other structural dimensions from a configuration-driven perspective and design the indicators to quantify the outcomes and performance of sustainability.

**Author Contributions:** Y.C. mainly wrote the original manuscript and summarized the literature and deduced the hypotheses. Then she was in charge of translation, data analysis and revision. Y.L. mainly constructed the theoretical framework and led the investigation, also in charge of the article revising guidance. J.B. was mainly involved in survey and investigation to collect original data and take part in the data preprocessing.

**Funding:** This research was funded by the Innovative Science Foundation (No. CYB18014) and Travel Talent Program (No. WMYC20181-028).

**Conflicts of Interest:** The authors declare no conflicts of interest.

**Appendix: Questionnaire on the Effect of Positional Embeddedness on Tour Supplier Integration**

**Tips**

Dear respondents:

How are you?

We are grateful for your precious time to fill out this version of questionnaire which aims to investigate how network embeddedness (mainly about the network characteristic of focal firm's position/location in its ego network where fulfilled with various ties among suppliers, customers, counterparts, government and some other organizations) impact on the efficiency of supply chain integration in tourism industry.

Meanwhile, it is also a self-examine, self-learning and self-improving process for surveyed company during answering questions and you can rethink how to enhance your core competitiveness effectively through collaboration, and then further recognize/understand the development of strategies. So, please answer the following questions seriously. If you do not know the explicit data, please try to estimate accurately. The reliability of your choice is of great importance.

We solemnly promise that the data collected by questionnaire is only for academic research, and would not be leaked. The investigation would not affect your business development. Therefore, we hope you give truthful answer. Sincerely thank you for your support and cooperation!

Best wishes!

**Appendix A**

**PART 1 Introduction**

1. Name: ____________ Job title: ____________
2. Telephone: ____________ Fax: ____________
3. E-mail: ____________________
4. Company name: ____________________
5. Address: ____________________
6. Postcode: ____________________

**Please type "√" or select the letter when you agree with the answer.**

7. The years of your firm's establishment: (   )

A. less than 5 years B. 5-10 years C. 11-20 years D. more than 20 years

8. Is your business a competitive relationship with other partners?    Yes (   ) No (   )

9. Could your core product/service be easily replaced by others?    Yes (   ) No (   )

10. The nature of your company: (     )

A. State-owned   B. Private   C. Sino-foreign joint   D. Foreign-invested   E. Others:____________

11. The field of your firm's core business: (     )

A. Online platform (OTA)   B. Offline service entity (travel agent)   C. Accommodation

D. Restaurant E. Entertainment F. Attractions G. Souvenir H. Transportation (airline/high-speed train/bus)   I. Others:____________

12. What resource could be provided by your firm during cooperation (     ), what resource could be provided by partners (     ), the core resources were (   )______(multiple choices)

A. Core digital resource   B. Knowledge about R&D on new product   C. IT resource

D. Consulting   E. Marketing (including distribution channels, sale skill, etc.)

F. Restaurant support   G. Accommodation support   H. Transportation support

I. Attraction support    J. Managerial skill    K. Human resource    L. Facilities

M. Investments N. Others__________

13. The sum of employees:

    A. less than 10    B. 11-50    C. 51-100    D. 100-500    E. 501-1000    F. more than 1000

14. How many companies cooperated with your company (  )

    A. 1-2    B.2-5    C.6-10    D.10-15    E.15-20    F. more than 20

15. The GMV (Gross Merchandise Volume) of your company per year is (  )

    A. Less than 0.5 million    B. 0.5-1 million    C. 1-5 million    D. 5-10 million    E. more than 10 million

16. Your position in operational department of your company:

    A. Chief executive.    B. Senior manager    C. Junior manager    D. Staff

17. Your working experience in your company:

    A. less than 2 years    B. 2-5 years    C. more than 6 years

## PART2 Tour supplier integration

Your attitude to the following description of your company's current supply chain integration could be measured by 1-5 Likert scale. 1 for disagree, 5 for agree, 2-4 for the increasing degree. Please use "√" to select your answer.

| 18. Tour supplier integration | Disagree | | | | Agree |
|---|---|---|---|---|---|
| 1.Our major supplier shares their service capacity with us frequently | 1 | 2 | 3 | 4 | 5 |
| 2.The establishment of quick ordering systems with major supplier | 1 | 2 | 3 | 4 | 5 |
| 3.The participation level of our major supplier is in the design stage | 1 | 2 | 3 | 4 | 5 |
| 4.Stable procurement through network with our major supplier | 1 | 2 | 3 | 4 | 5 |
| 5.We share our demand forecasts with our major supplier | 1 | 2 | 3 | 4 | 5 |
| 6.We help major supplier to improve its process to better meet our needs | 1 | 2 | 3 | 4 | 5 |
| 7.We Integrate the data systems with key suppliers | 1 | 2 | 3 | 4 | 5 |

**Appendix B**

## PART 3 Embeddedness

Please first check the list of enterprises provided by research team, and then select the enterprises having business contacts with your company. Type "√" on the choice you agree with.

Which organization would you contact through networks? (based on the offered organization list)

| Questions | TA1 | TA2 | TA3 | … … .. | TA51 | A1-4 | S1-4 |
|---|---|---|---|---|---|---|---|
| With whom often have a meal | | | | | | | |
| To whom transmit messages (email, WeChat..) | | | | | | | |
| With whom frequently contact | | | | | | | |
| With whom has close private relations | | | | | | | |
| With whom often cooperate deeply | | | | | | | |
| To whom often share the tourist flow | | | | | | | |
| Who to turn to when operating difficulties | | | | | | | |
| With whom found a study group | | | | | | | |
| With whom have joint investment | | | | | | | |
| with whom integrate tourism routes | | | | | | | |

**Appendix C  The Value of Positional Indicators in the TA Extended Network**

| | Dimensional Indexes of Positional Embeddedness | | | | | | | | | | |
|---|---|---|---|---|---|---|---|---|---|---|---|
| Respondent | Degree Centrality | Structural Hole | Closeness Centrality | Respondent | Degree Centrality | Structural Hole | Closeness Centrality | Respondent | Degree Centrality | Structural Hole | Closeness Centrality |
| TA1 | 0.466 | 0.468 | 0.65 | TA18 | 0.81 | 1.740 | 0.84 | TA35 | 0.534 | 0.436 | 0.68 |
| TA2 | 0.448 | 0.132 | 0.64 | TA19 | 0.707 | 1.269 | 0.77 | TA36 | 0.638 | 0.639 | 0.73 |
| TA3 | 0.483 | 0.155 | 0.66 | TA20 | 0.81 | 1.095 | 0.84 | TA37 | 0.621 | 0.569 | 0.72 |
| TA4 | 0.914 | 1.573 | 0.92 | TA21 | 0.828 | 2.343 | 0.85 | TA38 | 0.483 | 0.249 | 0.66 |
| TA5 | 0.776 | 0.941 | 0.82 | TA22 | 0.879 | 1.820 | 0.89 | TA39 | 0.517 | 0.438 | 0.67 |
| TA6 | 0.724 | 1.079 | 0.78 | TA23 | 0.828 | 1.902 | 0.85 | TA40 | 0.534 | 0.262 | 0.65 |
| TA7 | 0.724 | 0.682 | 0.78 | TA24 | 0.759 | 1.023 | 0.81 | TA41 | 0.466 | 0.620 | 0.68 |
| TA8 | 0.707 | 0.993 | 0.77 | TA25 | 0.759 | 0.736 | 0.78 | TA42 | 0.483 | 0.453 | 0.66 |
| TA9 | 0.828 | 1.237 | 0.85 | TA26 | 0.552 | 0.687 | 0.69 | TA43 | 0.724 | 0.994 | 0.78 |
| TA10 | 0.81 | 1.459 | 0.84 | TA27 | 0.448 | 0.266 | 0.64 | TA44 | 0.672 | 0.948 | 0.75 |
| TA11 | 0.759 | 1.591 | 0.81 | TA28 | 0.759 | 1.630 | 0.81 | TA45 | 0.466 | 0.770 | 0.65 |
| TA12 | 0.707 | 0.715 | 0.77 | TA29 | 0.759 | 1.654 | 0.81 | TA46 | 0.552 | 0.704 | 0.69 |
| TA13 | 0.672 | 1.094 | 0.75 | TA30 | 0.466 | 0.343 | 0.65 | TA47 | 0.741 | 0.998 | 0.79 |
| TA14 | 0.759 | 1.770 | 0.81 | TA31 | 0.569 | 0.203 | 0.7 | TA48 | 0.603 | 0.552 | 0.72 |
| TA15 | 0.793 | 0.833 | 0.83 | TA32 | 0.672 | 0.849 | 0.75 | TA49 | 0.69 | 1.356 | 0.76 |
| TA16 | 0.776 | 1.438 | 0.82 | TA33 | 0.707 | 1.007 | 0.77 | TA50 | 0.741 | 1.595 | 0.79 |
| TA17 | 0.828 | 1.556 | 0.85 | TA34 | 0.534 | 0.385 | 0.68 | TA51 | 0.586 | 0.863 | 0.71 |

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
