# Peer review of "Effect of Positional Embeddedness on Tour Supplier Integration: Chinese Evidence"

_sustainability, doi:10.3390/su11205741_

Round 1
Reviewer 1 Report
The paper deals with an interesting topic, which received little attention from the literature.
The theoretical background takes into account the relevant literature and the proposed research methodology is highly original.
I have some suggestion for the author, which - hopefully - could help to further strengthen the paper.
Keywords: in my opinion, “Sustainability” is too vague as a keyword. Maybe “Tourism sustainability” could sound better.
Introduction: In the last part of the Introduction, the author presents the purpose of the research. It should here be better explained how the research (and the research questions) deals with sustainability.
Theoretical background: In my opinion, lines 87-88 can be deleted.
Research methodology: the adequacy of the sample needs to be properly discussed. In my opinion, the adopted sample is one of the weaknesses of the research and this fact has also to be highlighted in the last section devoted to Limitation and further research.
Author Response
1,Keywords: in my opinion, “Sustainability” is too vague as a keyword. Maybe “Tourism sustainability” could sound better.
Thanks very much. Tourism sustainability is more specific. The key words have been renewed.
2,Introduction: In the last part of the Introduction, the author presents the purpose of the research. It should here be better explained how the research (and the research questions) deals with sustainability.
Thanks for your valuable advice. The research issues in this paper were focused on advocating the supplier development of sustainability (SDS) practices in the angle of travel agent by analyzing the resource integrating mechanism in order to upgrade the sustainability of TSC system. TAs as the intermediaries in TSC networks can ally suppliers with sustainability practices to integrate resources and can encourage the sustainable consumption choice in markets simultaneously to achieve the dual strategies and operating goals, finally to upgrade the sustainability performance of the whole TSC system. The introduction and conclusion have been renewed.
3,Theoretical background: In my opinion, lines 87-88 can be deleted.
Thanks for your advice. The sentences have been deleted.
4,Research methodology: the adequacy of the sample needs to be properly discussed. In my opinion, the adopted sample is one of the weaknesses of the research and this fact has also to be highlighted in the last section devoted to Limitation and further research.
The research conclusions were considered more helpful to the development of tourism in Chongqing (China). We hope the research methodology will be useful for other research area in tourism. Our research team will keep improving the research. SNA is very useful when analyzing the structure of relations among multiple stakeholders in tourism. We believe that there will be more valuable research finds based on SNA in future.
Reviewer 2 Report
The study is theoretically well founded and the hypotheses are consistent with the current state of the question. The method is appropriate and provides new ways of approaching the subject, although I must admit that reading is not easy if the reader has not previously worked with random permutations. On the other hand, when a cross-sectional design is used there is always doubt about the stability of the attributions, I understand that they should have insisted more on this topic by pointing out the limitations of the study. However, in my opinion, the conclusions are in line with the data found.
Author Response
The study is theoretically well founded and the hypotheses are consistent with the current state of the question. The method is appropriate and provides new ways of approaching the subject, although I must admit that reading is not easy if the reader has not previously worked with random permutations. On the other hand, when a cross-sectional design is used there is always doubt about the stability of the attributions, I understand that they should have insisted more on this topic by pointing out the limitations of the study. However, in my opinion, the conclusions are in line with the data found.
Thank you for your valuable advice, we will continue to work hard and pay more attention to the stability in the next research to achieve continuous improvement.
Reviewer 3 Report
The third hypothesis is based, in the case of the first hypothesis(3.1.),
on works that are older than 10 years.
Similar to the second hypothesis(3.2. & 3.3.).
I suggest trying to look for more recent points of view.
Author Response
Thank you very much for your valuable comments, which are very useful. After analyzing recent literature, we have added recent research findings which support the deducing logic of hypotheses more. The hypotheses (in part 3) have been renewed. Thank you very much! ~